# Non-Linear Dynamics Analysis of Protein Sequences. *Application to CYP450*

**DOI:** 10.3390/e21090852

**Published:** 2019-08-31

**Authors:** Xavier F. Cadet, Reda Dehak, Sang Peter Chin, Miloud Bessafi

**Affiliations:** 1PEACCEL, Protein Engineering Accelerator, 6 square Albin Cachot, box 42, 75013 Paris, France; 2LSE laboratory, EPITA, Paris 94276, France; 3Learning Intelligence Signal Processing Group, Department of Computer Science, Boston University, Boston, MA 02215, USA; 4LE2P-Energy Lab, Laboratory of Energy, Electronics and Processes EA 4079, Faculty of Sciences and Technology, University of La Reunion, 97444 St Denis CEDEX, France

**Keywords:** power law, Brownian process, Kolmogorov complexity, entropy, chaos, monofractal, non-linear, cumulative sum, sequence analysis, protein engineering

## Abstract

The nature of changes involved in crossed-sequence scale and inner-sequence scale is very challenging in protein biology. This study is a new attempt to assess with a phenomenological approach the non-stationary and nonlinear fluctuation of changes encountered in protein sequence. We have computed fluctuations from an encoded amino acid index dataset using cumulative sum technique and extracted the departure from the linear trend found in each protein sequence. For inner-sequence analysis, we found that the fluctuations of changes statistically follow a −5/3 Kolmogorov power and behave like an incremental Brownian process. The pattern of the changes in the inner sequence seems to be monofractal in essence and to be bounded between Hurst exponent [1/3,1/2] range, which respectively corresponds to the Kolmogorov and Brownian monofractal process. In addition, the changes in the inner sequence exhibit moderate complexity and chaos, which seems to be coherent with the monofractal and stochastic process highlighted previously in the study. The crossed-sequence changes analysis was achieved using an external parameter, which is the activity available for each protein sequence, and some results obtained for the inner sequence, specifically the drift and Kolmogorov complexity spectrum. We found a significant linear relationship between activity changes and drift changes, and also between activity and Kolmogorov complexity. An analysis of the mean square displacement of trajectories in the bivariate space (drift, activity) and (Kolmogorov complexity spectrum, activity) seems to present a superdiffusive law with a 1.6 power law value.

## 1. Introduction

From the information viewpoint, a protein sequence can be considered as a distribution of successive symbols extracted with a rule from a dictionary. Conceptually, it means that the protein sequence is simply encoded to a set of symbol combinations. Moreover, the number of the symbols used is usually very small in comparison to the length of the protein sequence. Consequently, there is a huge variety of combinations of symbols to encode a protein sequence in the real world. It is well-known that the molecular mechanism (stability, structure function, disorder) is often triggered by complex interactions [1,2,3]. Like the emerged part of an iceberg, the intricated symbol set of an encoded protein sequence can be seen as a footprint of a wide range of covert biochemical interactions within the protein. Then, there are numerous encoder models that try to reflect the reality accurately using a conversion rule related to physicochemical and biochemical properties [4,5,6]. Beyond the symbol combination and arrangement of the protein sequence, understanding the nature and the organization of the symbols is very challenging in protein biology. Therefore, analyzing the encoded protein sequence by means of nonlinear analysis can provide some insights about the dynamics of the changes within the dataset. Searching for similarities between encoded protein sequences in a dataset is one of the important advantages of morphological analysis of protein sequences. There are many approaches to extract groups, which are conceptually based on a clustering method of global or local information about the protein sequence [7,8,9,10,11,12,13]. The prediction of disorder of the protein sequence is often related to the ability to track the degree of randomness, the stochasticity, and the complexity embedded in the whole encoded dataset. There are studies which focus on randomness, chaos, long-range interaction between sequences for classification, and predictability. For example, Yu et al. [14] have made a comparative study of structure and intrinsic disorder between 10,000 natural and random protein sequences and found that natural sequences have more long disordered regions than random sequences. In addition, Gök et al. [5] have used the Lyapunov exponent and test four classifier algorithms (Bayesian network, Naïve Bayes, k-means, and SVM) to identify the disordered protein regions. Long short-term memory (LSTM) recurrent neural networks is a deep learning algorithm that has gained some interest for tracking the long-range interactions between sequences [1,15]. These studies reveal that there is potential information about degree of randomness, disorder, and stochasticity in protein sequences and beyond some degree of predictability. It means that the protein sequence exhibits some order within disorder and changes are not a likelihood for this set of symbols. To find out what kind of information and properties of disorder or complexity we are able to extract from protein sequences, we propose to scan the changes inside the protein sequences and between sequences using a multidisciplinary approach. It means that we intend, at the same time, to use tools from information theory field (entropy of information, Kolmogorov complexity), physical theory (chaos, fractional Brownian processes, drift-diffusion processes), and signal processing (multifractality, Fourier analysis). To our knowledge, the use of multidisciplinary tools to analyze the dynamics of the changes within a protein sequence and between sequences is new. As mentioned previously, the encoded protein sequence contains successive numerical values and can also be considered as a time series. The aim of this paper is to encompass the variability of the inner changes hidden behind the encoded protein sequence using nonlinear tools, and to assess the predictability of the underlying non-stationary protein sequence activity.

The study is organized as follows. Section 2 presents the experimental dataset and the encoded protein sequence. Section 3 describes the algorithm used to analyze the time series (i) entropy and chaos, (ii) Kolmogorov complexity and Turing machine, (iii) law-scaling and stochastic process, and (iv) surrogated and shuffled data. Finally, Section 4 includes both presentation of the results obtained and discussion. The concluding remarks are given in Section 5.

## 2. Experimental Dataset

To facilitate the understanding of readers outside the realm of life sciences, we will provide a brief definition of a polypeptide/protein sequence. A protein sequence is a chain made of residues of amino acids. Twenty amino acids are the basic building blocks for proteins. We will provide an application example as well.

### 2.1. Alphabetical Dictionary

Each amino acid is represented by a letter corresponding to the one-letter code for an amino acid. The global sequence has a biological meaning. A single variation in the sequence could have a huge impact on the activity of the protein. An example of a protein sequence (Cytochrome P450) is given below:

MTIKEMPQPKTFGELKNLPLLNTDKPVQALMKIADELGEIFKFEAPGRVTRYLSSQRLIKEACDESRFDKNLSQALKFVRDFAGDGLATSWTHEKNWKKAHNILLPSFSQQAMKGYHAMMVDIATQLIQKWSRLNPNEEIDVADDMTRLTLDTIGLCGFNYRFNSFYRDSQHPFITSMLRALKEAMNQSKRLLRLWPTAPAFSLYAKEDTVLGGEYPLEKGDELMVLIPQLHRDKTIWGDDVEEFRPERFENPSAIPQHAFKPFGNGQRACIGQQFALHEATLVLGMILKYFTLIDHENYELDIKQTLTLKPGDFHISVQSRHQEAIHADVQAAE

### 2.2. An Application Example: Cytochrome P450

Cytochrome P450 is a protein, i.e., a polypeptidic sequence of 464 or 466 amino acids. It is used to generate products of significant medical and industrial importance. Three parental cytochromes P450, i.e., CYP102A1(P1), CYP102A2(P2), and CYP102A3(P3) were used to generate 242 chimeric sequences of cytochrome P450 [16]. Further, 242 thermostable protein sequences were created by recombination of stabilizing fragments. For each variant, the thermostability (defined herewith as: Activity) was analyzed by the measurement of the T_50,_ T_50_ being the temperature at which 50% of the protein was irreversibly denatured after incubation for 10 min. The result is a decrease in activity. Activity ranges from 39.2 °C to 64.48 °C. Chimeras are written according to fragment composition: 23121321 represents a protein that inherits the first fragment from parent P2, the second from P3, the third from P1, and so on.

## 3. Methodology

In this study, the questions are: “*Can statistical, nonlinear, and complexity analysis give us some information about the pattern in a protein sequence and its changes along the sequence and also the next, or other sequences? Can we group sequences according to their activity but also their morphological pattern?*”. To assess the ability of the statistical chaos and complexity tools, we have transformed each protein sequence into numerical or binary time series according to the need of the use of the tool.

First of all, there exist different conversion tables to transform protein residues (letters) to numerical sequences. We have used the freely available one, namely AA index database [17,18]. This database contains a huge number of ascribed numerical values for each protein residue. There are 566 numerical values, which are for each index in the sequence univocally in correspondence with physicochemical and biochemical properties of the residues. In this case, we have selected the index 532 in the dataset, which allows us to rank and encode 20 standard amino acids.

### 3.1. Entropy and Chaos

Entropy is a concept that was first discovered in physics. Nevertheless, this concept is also encountered in other fields and especially in the theory of information. In 1948, Shannon [19] formalized the concept of entropy of the information H of a string of length N, which contains Q repeated symbols $S=\left\{s_{1},\;s_{2},\;\ldots ,\;s_{Q}\right\}$. H is shown by the well-known formula:(1)$$H=-\sum_{i=1}^{Q}{\hat{p}}_{i}log{\hat{p}}_{i}$$
where ${\hat{p}}_{i}=\frac{N_{s_{i}}}{N}$.

$N_{s_{i}}$ is the number of appearances of the symbol $s_{i}$ in the string of length N. Thus, $p_{i}$ is the probability of occurrence within the range value ]0 1]. As we suppose that all Q symbols exist in the string, the probability 0 is excluded. The minus sign is to ensure a positive value of the entropy H as the logarithm is always negative. H is a global measure of the total amount of information in an entire probability distribution contained in a sequence.

Another measure of entropy is the sample entropy [20]. Let us consider a set of N symbols $s_{i,k}$ in a sequence $S_{i}$ chosen among M sequences in the dataset. From the sequence $S_{i}$ we extract two subsets of m symbols $S_{i,p}^{m}=\left\{s_{i,p},\;s_{i,p+1},\ldots ,s_{i,p+m}\right\}$ and $S_{iq}^{m}=\left\{s_{i,q},\;s_{i,q+1},\ldots ,s_{i,q+m}\right\}$ where p≠q. The parameters p and q correspond to the index position of the first symbol of respectively the subset $S_{i,p}^{m}$ and $S_{i,q}^{m}$ within the sequence $S_{i}$. The sample entropy (SampEn) of the sequence $S_{i}$ is defined as $SampEn{\left(m,r,N\right)}_{i}=-log\left(\frac{A_{i}}{B_{i}}\right)$, where $A_{i}$ is the number of pair-wise subset symbols $\left(s_{ip}^{m+1},s_{iq}^{m+1}\right)$ of length m+1 with a distance $d\left(s_{ip}^{m+1},s_{iq}^{m+1}\right)<r$ while $B_{i}$ is the number of pair-wise subset symbols $\left(s_{ip}^{m},s_{iq}^{m}\right)$ of length m with a distance $d\left(s_{ip}^{m},s_{iq}^{m}\right)<r$. The r is a threshold value of similarity between the pair-wise subset symbols $\left(s_{ip}^{m},s_{iq}^{m}\right)$. In our study, the sequence is a set of numbers. Then, the distance $d\left(s_{ip}^{m},s_{iq}^{m}\right)$ is a Euclidian distance and the tolerance value threshold value r is chosen between 0.1 and 0.2 of the standard deviation of the sequence $S_{i}$ [20]. Moreover, the embedding dimension m is usually taken to be 2. Finally, the sample entropy is a positive value, which can be 0 for a regular sequence and roughly 2.2 or 2.3 for a strongly irregular sequence. The sample entropy is a measure of the regularity within a sequence.

In addition, sometimes an irregularity pattern in a time series could be related to the chaos process within a sequence. The largest Lyapunov exponent is the most common parameter used to characterize chaos in a dynamical system. The sign and the value of this parameter give an indication of the response of a system to amplify, damp, or oscillate a small perturbation. In our case, it means that if the largest Lyapunov exponent is (i) positive, then the process is chaotic, (ii) close to zero, then the process is periodic or quasi-periodic, and finally (iii) negative, the process is damping and has an attractor. In our study, to achieve the search for chaos pattern in a sequence $S_{i}$, we have used Wolf’s algorithm [21] to compute the Lyapunov exponent spectrum and the largest Lyapunov exponent (LLE).

### 3.2. Kolmogorov Complexity and Turing Machine

Let us assume we have a set of M sequences $S=\left\{S_{1},\;S_{2},\;\ldots ,\;S_{M}\right\}$. Then, we suppose that we have for each sequence i of string $S_{i}$, a set of N values defined as $S_{i}=\left\{{p_{i}}^{1},\;{p_{i}}^{2},\;\ldots ,\;{p_{i}}^{N}\right\}$. To assess disorder within a sequence, we use the Kolmogorov complexity method [22]. This method is based on the concept of Turing machine and the mathematical expression of the algorithmic complexity can be written $K_{T}\left(s\right)=min\left\{\left|p\right|,\;T\left(p\right)=s\right\}$. This states that the algorithmic complexity of a string s is the shortest program p computed with a Turing’s machine T to gather output s [23,24]. To compute the Kolmogorov complexity (KC), there are three processes: (i) Convert the sequence $S_{i}$ to binary sequence $B_{i}$ using a threshold method, (ii) compress the sequence $B_{i}$ with Lempel-Ziv compressor to a compressed sequence $C_{i}$, and (iii) compute and normalize the Kolmogorov complexity number associated with the original sequence $S_{i}$.

Binarizing the sequence $S_{i}$ is based on the particular value used as threshold value ${p_{i}}^{T}$ to assign each number ${p_{i}}^{k}$ in the sequence $S_{i}$ with the value of 0 if ${p_{i}}^{k}$ is less than the threshold value ${p_{i}}^{T}$, or conversely assigned with the value of 1 if ${p_{i}}^{k}$ exceeds the threshold value ${p_{i}}^{T}.$ The mathematical expression of the binary value of the number ${p_{i}}^{k}$ in the sequence $S_{i}$ is:
(2)$${B_{i}^{k}|}_{\begin{array}{c} i=\left\{1,2,\;\ldots ,\;M\right\}\\ k=\left\{1,2,\;\ldots ,\;N\right\} \end{array}}=\{\begin{array}{ccc} & 0 & if\;{p_{i}}^{k}<{p_{i}}^{T}\\ or & & \\ & 1 & if\;{p_{i}}^{k}\geq {p_{i}}^{T} \end{array}$$
where ${p_{i}}^{T}$ is a threshold value of sequence $S_{i}$.

Usually, the mean of the set $\left\{{p_{i}}^{1},\;{p_{i}}^{2},\;\ldots ,\;{p_{i}}^{N}\right\}$ is used as a threshold value of the sequence $S_{i}$. Nevertheless, we will take into account the amplitude of the numbers ${p_{i}}^{k}$ to compute the optimum threshold value $p_{i}{{}^{T}}_{opt}$ associated with the sequence $S_{i}$. Thus, we introduce the Kolmogorov complexity spectrum (KCS), which is an iterative procedure to compute the Kolmogorov complexity for various threshold values within the range values ${p_{i}}^{k}$ of the sequence $S_{i}$ [25]. The encoding number to binary value is presented as:
(3)$${B_{i}^{k}|}_{m}=\{\begin{array}{ccc} & 0 & if\;{p_{i}}^{k}<{p_{i}}^{T_{m}}\\ or & & \\ & 1 & if\;{p_{i}}^{k}\geq {p_{i}}^{T_{m}} \end{array}\;m=1,2,\ldots ,K$$
where ${p_{i}}^{T_{L}}=min_{k}\left(\left\{{p_{i}}^{k}\right\}\right)+m\left\{\frac{max_{k}\left(\left\{{p_{i}}^{k}\right\}\right)-min_{k}\left(\left\{{p_{i}}^{k}\right\}\right)}{K-1}\right\}$.

Thus, for each sequence $S_{i}$, the Kolmogorov complexity spectrum is a set of K Kolmogorov complexity values $K{C_{i}}^{K}=\left\{K{C_{i}}^{1},\;K{C_{i}}^{2},\;\ldots ,\;K{C_{i}}^{K}\right\}$. The optimum threshold $p_{i}{{}^{T}}_{opt}$ is chosen among the set of threshold values $\left\{{p_{i}}^{T_{1}},\;{p_{i}}^{T_{2}},\;\ldots ,\;{p_{i}}^{T_{K}}\right\}$ using the condition $p_{i}{{}^{T}}_{opt}=\left\{{p_{i}}^{T_{j}}\;|\;K{C_{i}}^{j}=max_{k}\left(K{C_{i}}^{k}\right)\right\}$.

The compression method used in this study is the Lempel-Ziv compressor [26]. This is an iterative search in the binary series $B_{i}$ of the overall possible subset sequences, which are different from each other. The result is a compressed sequence $C_{i}$. If $\left|C_{i}\right|$ represents the length of the compressed binary sequence $C_{i}$, then Kolmogorov complexity $KC_{i}$ associated with the sequence $S_{i}$ is:
(4)$$KC_{i}=\left|C_{i}\right|log_{2}N/N.$$

The term $log_{2}N/N$ in the expression of $KC_{i}$ insures the normalization of the Kolmogorov complexity.

### 3.3. Law-Scaling and Stochastic Process

As previously mentioned, a sequence is defined as a set of alphabetic letters, which could be converted to other symbols (numerical, binary, etc.). Nevertheless, the changes of symbols along the chain are usually related to the real world of biochemical activities along the protein sequence. The question is “*Do those changes present a regular or irregular pattern within a sequence which can provide some information about an underlying dynamic in a sequence?*” First, we have to define the changes in a sequence i of pairwise symbols separated by a distance, namely an increment of position. Let us assume d is the increment pairwise symbols and the quantity $\Delta {p_{d}}_{i}={\left|{p_{i}}^{j}-{p_{i}}^{k}\right|}_{d=\left|k-j\right|}$ is the magnitude of changes of the pairwise symbols separated by an increment of d. We define the structure function ${S_{q}}_{i}\left(d\right)$ for a sequence i defined by the expression ${S_{q}}_{i}\left(d\right)=\frac{1}{{N_{d}}_{i}}\sum_{m=1}^{{N_{d}}_{i}}{\left|{p_{i}}^{j}-{p_{i}}^{k}\right|}_{d=\left|k-j\right|}^{q}$ where ${N_{d}}_{i}$ is the number of pairwise symbols separated with a distance d. By extension, this function can also be used to track the existence of scaling law in the data ${S_{q}}_{i}\left(d\right)\propto d^{\xi \left(q\right)}$. ξ(q) is the generalized Hurst exponent, which is indicative of the nature of pairwise symbol changes and the stochasticity of processes like long-term memories, Brownian motion, self-similarity pattern [27]. The probability function (PDFs) of the distribution of the normalized changes of pair-wise symbols $\Delta {p_{d}}_{i}/\sigma \left(\Delta {p_{d}}_{i}\right)$ within a sequence i can be computed to analyze the normality of the changes in a sequence. Additionally, kurtosis or flatness is another measure of the normality of the changes of the pairwise symbols. For sequence i, the kurtosis $F_{i}=\;S_{4}{}_{i}\left(d\right)/{\left(S_{2}{}_{i}\left(d\right)\right)}^{2}$. The terms ${S_{4}}_{i}\left(d\right)$ and ${S_{2}}_{i}\left(d\right)$ are, respectively, the fourth- and second-order moment of the pairwise distribution.

### 3.4. Surrogated and Shuffled Data

The methods to surrogate and shuffle the data are very popular tools to assess the existence of nonlinearities and the scaling properties of a process. Both algorithms are based on the generation of randomized synthetic data using specific constraint rule to generate the synthetic data. Surrogated data used in this study are the iterative amplitude-adjusted Fourier transform (IAAFT). This method preserves the statistical properties of the original data but randomizes the phase spectrum of the Fourier transform of the original data. The synthetic data generated with this method lead to removing nonlinearities in the original data. Shuffled data are obtained by a random permutation between values of the original data. This method is a bootstrapping algorithm without repetition of the indices’ permutation. Variants of the protein (synthetic sequences) are obtained by variation of any position in the sequence and not by variation of the fragments constitutive of the protein (described in the Section 2.2 “An application example: Cytochrome P450”). The data obtained are a set of values that do not exhibit any linear correlation in the synthetic data and preserve the amplitude distribution. For more information about these two algorithms, the reader can refer to the review of Schreiber and Schmitz [28].

## 4. Normalized Detrended Cumulative Sum (NDCS) Method

Fluctuations or changes along the protein sequence are of interest in this study but we need to show how we extract this information from the original data. Cumulative sum is a sequential method that is widely used to detect changes in a time series and to track the self-similarity in a dataset [29]. In this study, we have applied this algorithm for each sequence and generated a new sequence of fluctuations defined as a departure from the linear trend. Within the 242 protein sequences of a length of 466 for each one, each index in a sequence is originally labelled with an alphabetical letter. There are 20 letters used (A, C, D, E, F, G, H, I, K, L, M, N, P, Q, R, S, T, V, W, and Y) corresponding to the one-letter code for amino acid. In this study, the D PRIFT index is chosen from the AA index catalog to convert the alphabetical symbols to numerical values [30]. It allows us to distinguish each of the 20 amino acid residues by a unique value related to its hydrophobicity property. The encoding process, which converts the original alphabetical letters to numerical values within the [−5.68 6.81] range, is shown in Table 1.

We are aware that this description by their hydrophobicity values is oversimplified and does not account (i) for many other properties of amino acids that are well known to strongly affect pattern changes in protein sequences along families, such as volume, aromaticity, and different charge states for the same amino acid in distinct positions or, (ii) for the fact that the exposure of continuous amino acids sequences to solvent or their occlusion in protein cores is a fundamental requirement for proteins to fold in functional arrangements, giving importance to hydrophobic and polar amino acids and their distribution. However, whatever the choice among all the possible amino acid indexes that are able to distinguish between the 20 amino acid residues, the index will be insufficient.

As shown in Figure 1a, the distribution values show a non-normal distribution, which is indicative of the non-gaussian process along the protein sequence. Roughly, the distribution looks like a U-shape where the highest probability of occurrence is obtained for the extreme values and the lowest for the mean value of the available D PRIFT index. Then, the pattern of the encoded protein sequence appears like complex bounced stairs with randomness as a sharp jump (Figure 1b).

To target the jump stair pattern analysis within the protein sequence, we have used the normalized detrended cumulative sum (NDCS) method. The cumulative sum is a well-known and widely used algorithm to detect changes and shifts in time series [31]. In this study, we have extracted the linear long-term and normalized the cumulative sum of each sequence to (i) focus on the local change and (ii) have the same scale to compare transformed data. Figure 2 presents an example of transforming the original data (Sequence 1) into a detrended cumulative sum data. For clarity, we only present here the cumulative sum and linear detrending of the data. The normalized process is shown in the next figure. The trend of the cumulative sum is considered to be a linear trend for all the 242 protein sequences. The negative drift of the cumulative sum is related to the mean of a sequence. In our dataset, the average of the D PRIFT index is negative for each sequence and explains the downward drift of the cumulative sum.

Figure 3a depicts the NDSC plot in comparison with the original data (Sequence 1). Fluctuations reflect the local changes along the sequence and also a significant change pattern around the middle of the sequence. The fluctuation pattern relying on the cumulative sum transformation involves continuous distribution, conversely to the discrete distribution of the original D PRIFT index (Figure 3b).

Figure 4a shows that the fluctuations of the NDCS of the D PRIFT index changes are normally distributed, with skewness close to 0 and kurtosis close to 3, which are the expected values for a normal distribution. In addition, the QQ-plot displayed in Figure 4b reveals that the observed distribution is close to a normal distribution and the two samples’ (dataset values and generated normal data values) Kolmogorov–Smirnov test applied to this distribution does not reject the null hypothesis at the 5% significance level.

## 5. Results and Discussion

### 5.1. Normality and Intermittency

The changes along the protein sequence for four different pairwise distances show a platykurtic nature (Figure 5a). The average distribution exhibits large amplitude for fluctuations greater than 2.5 times the standard deviation of NDCS of D PRIFT index changes. The average is computed using 242 protein sequences. Below this threshold value, the distribution is close to the Gaussian distribution. This kind of departure from the Gaussian distribution in fluctuations is indicative of intermittency. Moreover, Figure 5b highlights that the platykurtic nature of the fluctuations covers a wide range of pairwise distances, but it is more pronounced with the [30–60] pairwise distance and for distances less than 10 pairwise. To summarize, this flat distribution indicates more diversity of changes for the large amplitude of pairwise distance within the protein sequence.

### 5.2. Kolmogorov’s Law and Brownian Process

We have conducted a Fourier analysis to focus on the fluctuation of the NDCS of D PRIFT index changes. Surprisingly, scale invariance can be detected in the log-log presentation of the Fourier spectra (Figure 6a). An average of −1.68 based on power law is obtained, which is very close to the Kolmogorov power law result of −5/3. This highlights that the fluctuations of the NDCS of D PRIFT index changes along a sequence are similar to a non-stationary process and obey the famous Kolmogorov’s law of the energy cascade for turbulence in the inertial scale range [22]. In addition, as shown in Figure 6b, the range scale value for each sequence is rather close to −5/3, with an observed minimum slope value of −1.56 and a maximum slope value of −1.84. This means that the changes within the protein sequence can be formulated according to Fourier transform as $E\left(f\right)=f^{\beta }$ where β is the slope of the law and is close to the Kolmogorov spectrum. In addition, we can use criteria to check if the changes of protein are stationary or not [32]. This is summarized by the following test: β<1, the changes are stationary,β>1, the changes are non-stationary,1<β<3, the changes are non-stationary with stationary increments.

Thus, the changes in the sequence protein follow a non-stationary process. Moreover, the coefficient of variation of the fluctuations of the NDCS of D PRIFT index changes computed for all 242 sequences is less than 3%, confirming that this similarity with the Kolmogorov spectrum seems to be reproducible for each protein sequence as confirmed by the distribution of the spectrum slope obtained randomly with surrogated and shuffled data.

As shown previously in Figure 3b, the fluctuations of the NDCS of D PRIFT index changes appear to show seemingly organized fluctuations. The question is “*Is there some dynamic pattern of these change fluctuations along a sequence*
$S_{i}$
*and is there some randomness of changes within the protein sequence?*”. A first approach is to analyze the behavior of the fluctuation of the pairwise protein index. Figure 7a shows that on average, the second-order moment ${S_{2}}_{i}\left(d\right)$ of the pairwise protein sequence index separated by a distance d is linearly scaled in a sequence between pairwise protein sequence indexes separated by a distance d roughly below 50. We found a power law of 0.87, which is close to the Brownian power law process. Then, the behavior of the change fluctuations along each protein sequence $S_{i}$ seems to be close to a Brownian process. Furthermore, we found for each protein sequence a power law between a range of [0.69 0.99] and a coefficient of variation less than 7%, which reveals that the fluctuations of NDCS of the D PRFIT index changes along a sequence $S_{i}$ statistically have a behavior close to a Brownian process in regard to the results obtained with the surrogated and shuffled data (Figure 7b).

In addition, we have also computed the *q-order* moment for each protein sequence $S_{i}$. The result is shown in Figure 8a. As observed with second-order moment ${S_{2}}_{i}\left(d\right)$ analysis, we again have a scaling law distribution between pairwise protein sequence index $S_{i}$ below d=50 for a higher-order moment. This result reveals the existence of a monofractal feature along the protein sequence $S_{i}$. Figure 8b shows that the fluctuations of NDCS of D PRIFT index changes of each protein sequence $S_{i}$ contain a monofractal feature with ξ(q)=0.43 q, which is a linear law of q and reveals monofractal behavior. The slope of the linear law is called the Hurst exponent H. As a reminder, if the value of $H=\frac{1}{2}$, it means the changes in a sequence contain no memory as for the Brownian motion. If the changes of the sequence are anti-persistent $\left(0<H<\frac{1}{2}\right)$, then the main pattern of the changes shows that a decrease is followed by an increase and vice-versa. Finally, if the Hurst exponent is as $\frac{1}{2}<H<1$, then there is a persistent behavior in the changes and an increase or decrease will be maintained in a sequence. In our case, the changes are anti-persistent and they are statistically embedded between Kolmogorov process $\xi \left(q\right)=\;\frac{q}{3}$ [22] and the Brownian process $\xi \left(q\right)=\;\frac{q}{2}$. Thus, there is a potential stochastic model like the fractional Brownian model to predict the changes along the protein sequence.

### 5.3. Entropy, Chaos, and Complexity

As previously mentioned, a sequence is defined as a set of alphabetic letters, which could be converted to other symbols (numerical, binary, etc.). Nevertheless, the changes of symbols or numerical values along the sequence are usually related to the real world of biochemical activities inside the whole protein sequence. The question is “*Do those changes present regular, irregular, chaotic and complex pattern within a sequence*?” Furthermore, nonlinear analysis is one approach to estimate the changes in features along a sequence. In this study, we have used five algorithms to assess the degree of the randomness or the disorder and complexity in protein sequences: (i) The Shannon entropy (ShEn); (ii) the sample entropy (SampEn); (iii) the largest Lyapunov exponent (LLE); (iv) Kolmogorov complexity (KC); and (v) the Kolmogorov complexity spectrum (KCS) algorithm. Table 2 presents the descriptive statistics of the NDCS of D PRIFT index changes for 242 protein sequences. On average, there is a significant amount of information in an entire probability distribution contained in a sequence. We observe that SampEn and LLE values are close to one. Moreover, the KC method underestimates the complexity in comparison to the KCS method, which takes into account the amplitude of the changes. Following the comparison with the surrogated and shuffled data generated from the original data, we found that the NDCS of D PRIFT index changes for 242 protein sequences used in this study include stochastic and moderate chaotic processes and show apparent embedding between the Kolmogorov (H=1/3) and Brownian (H=1/2) monofractal processes.

### 5.4. Drift (DRF), Kolmogorov Complexity Spectrum (KCS), and Activity (ACT): Linear Correlation and Superdiffusive Process between Sequences

The activity as defined in Section 2.2 (Thermostability) is also freely available for each protein sequence. Figure 9a shows the cumulative sum of activity, entropy, chaos, complexity, fractal, and drift parameters for 242 protein sequences. In order to track the biochemical activity changes through an invariant sequence arrangement, we have sorted, in ascending order, each sequence with increasing activity. Then, we have also sorted the remaining parameters in respect to the increasing activity and applied the cumulative sum. For clarity, we have presented the 10th of the entropy, chaos, complexity, fractal, and drift parameters, and the 1000th for activity. Most of the curves show a slightly linear shape, which is the average mode through increasing sequence activity. Nevertheless, the dynamic of changes through this increasing activity highlights that NDCS’s activity changes are well correlated with the NDCS of Kolmogorov complexity spectrum and drift (Figure 9b). There are pronounced parabola with an open upwards shape for activity (ACT) changes and a conversely open downwards shape for the Kolmogorov complexity spectrum (KCS) and drift (DRF) changes. The correlation coefficient is very high between ACT, KCS, and DRF as shown in Figure 9c.

We found a relationship between the inner-sequence changes drift, the complexity, and the activity throughout crossed 242 rearranged increasing activity protein sequences. As shown in Figure 9c, the trajectories of the bivariate parameter (drift, activity) or (complexity, activity) exhibits trajectories with jump between sequences, which leads to the question: “*Are these successive jumps related to variable changes ruled by a power law?*”. Then, we have analyzed these trajectories by calculating the mean square displacement of changes $\langle {\left(\Delta d_{S}\right)}^{2}\rangle$ in the bivariate parameter (drift, activity) or (complexity, activity) space where $d_{S}$ is the distance between two sequences. Moreover, we defined the mean square displacement as $\langle \Delta {\left(d_{S}\right)}^{2}\rangle =\frac{1}{{N_{d}}_{S}}\sum_{m=1}^{{N_{d}}_{S}}{\left[{\left(X^{j}-X^{k}\right)}^{2}+{\left(ACT^{j}-ACT^{k}\right)}^{2}\right]}_{d_{S}=\left|k-j\right|}$ where ${N_{d}}_{S}$ is the number of pairwise sequences separated by a distance $d_{S}$ and X is the drift (DRF) or Kolmogorov complexity spectrum (KCS). Figure 10 shows $\langle \Delta {\left(d_{S}\right)}^{2}\rangle ∼{d_{S}}^{\alpha }$ with α∼1.7 for the drift and α∼1.6 for the complexity. We found that there is a scaling law of the bivariate (DFT, ACT) or (KCS, ACT) parameter that is similar to a super diffusive process with an exponent coefficient α>1 [33]. Here, we have plotted $\langle \Delta {\left(d_{S}\right)}^{2}\rangle /\langle \Delta {\left({d_{S}}_{c}\right)}^{2}\rangle$ where ${d_{S}}_{c}$ is the characteristic distance between two sequences computed with the correlation function $\langle \delta \left(d_{S}\right)\rangle =\frac{1}{{N_{d}}_{S}}\sum_{m=1}^{{N_{d}}_{S}}{\left[X^{j}X^{k}+ACT^{j}ACT^{k}\right]}_{d_{S}=\left|k-j\right|}$.

## 6. Conclusions

In this work, we analyze the nonlinear behavior of the D-PRIFT index changes around the overall linear trend scale of the protein sequence. To assess the nonlinear analysis, we have used protein residue values that are freely available, namely the AA index database. The protein dataset used contains 242 sequences and each sequence has 466 numerical values, one per amino acid residue. A protein sequence corresponds to a combination of encoding symbols from a dictionary of 20 standard amino acids symbols.

We have applied to each sequence a normalized detrended cumulative sum algorithm to extract the fluctuations of the numerical signal in the protein sequence. We analyzed these fluctuations with different tools, which are related to (i) entropy (information and regularity); (ii) chaos (largest Lyapunov exponent); (iii) complexity (Kolmogorov complexity and Kolmogorov complexity spectrum); and (iv) fractal (Hurst exponent). First, we showed that the change fluctuations of all the studied 242 protein sequences in the dataset seem to be non-stationary and follow on average a −5/3 Kolmogorov power-law. This result seems to be statistically significant in regard to a coefficient of variation less than 2% and a test done with randomly generated synthetically obtained data with surrogate and shuffle technique. To understand the nature of the inner changes within the protein sequence, we achieved the analysis of the variance of the changes through the scope of the spatial correlation: Here, the index position within the protein sequence. We found an invariance of pairwise scale index d, which is ruled by a ${S_{2}}_{i}\left(d\right)\propto d^{\alpha }$ with α=0.87, a coefficient close to one of the well-known stochastic Brownian processes. The dispersion of the slope obtained for all 242 protein sequences is statistically coherent in comparison with the results obtained with synthetic data. Following the local analysis of the changes along the protein sequence, we have performed a systematic q-order moment of the fluctuations in order to track if there is a self-similar repeating pattern in the inner sequence. We showed that change fluctuations within the protein sequence have a monofractal behavior, which is an average among the 242 sequences embedded between the Kolmogorov and Brownian monofractal processes with a Hurst exponent ranging between 1/3 and 1/2. To encompass the local analysis and to have an overview of the nonlinearity analysis, we have computed statistical parameters related to entropy, chaos, complexity, and fractality. We demonstrated that the NDCS of D PRIFT index changes for the 242 protein sequences used in this study exhibit statistically moderate complexity, and low chaotic fluctuations.

Moreover, to integrate these results in the analysis of the protein activity changes for each sequence, we have conducted a study of the relationship between the linear-trend (drift) computed with the cumulative sum algorithm, the Kolmogorov complexity spectrum, which is indicative of computational complexity, and the activity of each protein sequence. As this analysis focused on the dynamics of the changes, we also applied the normalized detrended cumulative sum for these three parameters as done for the inner-sequence analysis. The results show a strong linear relationship between the bivariate (drift, activity) and (complexity, activity) parameters, which provides insight into the potential use of drift and complexity as a predictor in a linear model. Moreover, the analysis of the trajectories in the bivariate space highlights superdiffusive behavior of the change fluctuations with a power-law around −1.6 of the mean square displacement for each chosen bivariate parameter. This study demonstrates that the changes in the inner sequence and throughout the crossed inter-sequence are nonstationary, stochastic, irregular, complex, weakly chaotic, and monofractal. To conclude, there is some predictability of protein sequence changes, which can be modelled using a stochastic model. Linear law and scale invariance features found in this study should be explored in future work to study for classification, regression predictive model, and could be useful in the field of protein engineering.

## Figures and Tables

**Figure 1 entropy-21-00852-f001:**
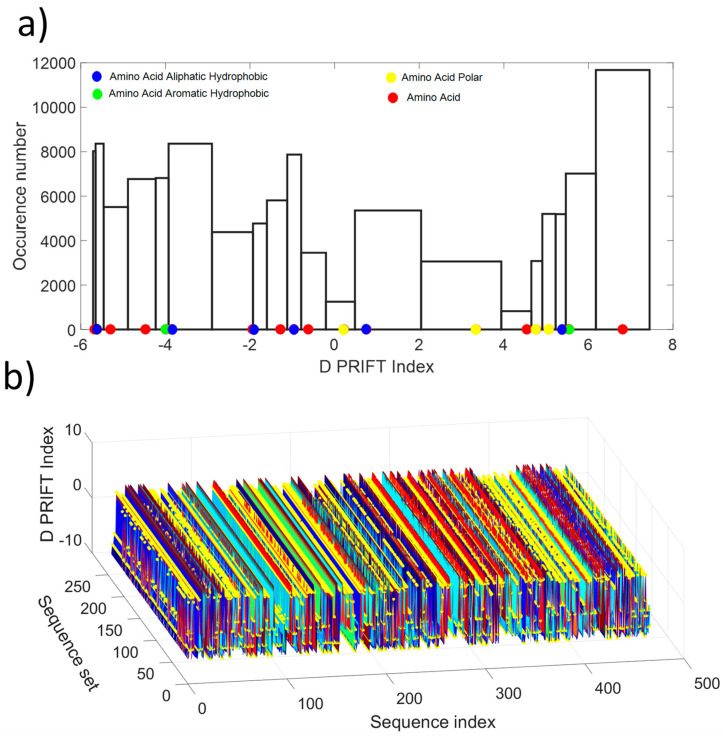
(**a**) Histogram of the D PRIFT index for 242 protein sequences. Red, blue, green, and yellow dots along the x-axis corresponds to the 20 values of the D PRIFT index. (**b**) Global view of the converted dataset (i.e., 242 protein sequences) using D PRIFT index rule. Yellow circle is indicative of the position within each sequence of the aliphatic hydrophobic, aromatic hydrophobic, and polar amino acids.

**Figure 2 entropy-21-00852-f002:**
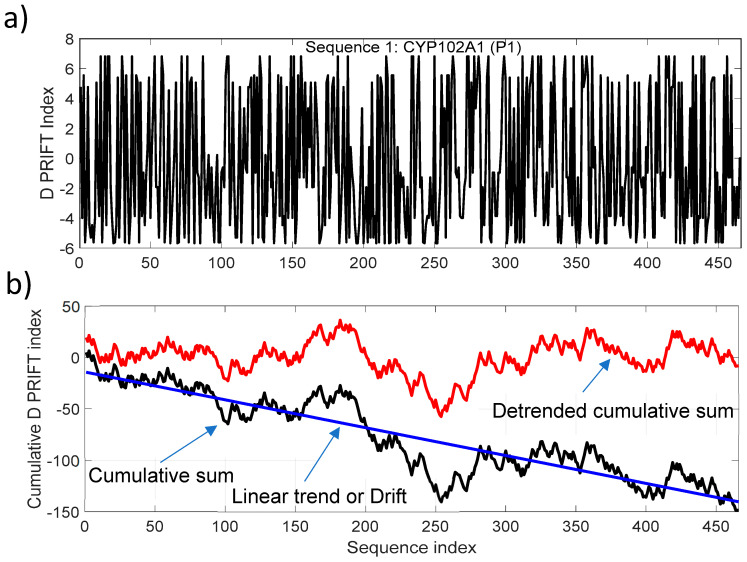
(**a**) D PRIFT index of sequence 1, which is parent CYP102A1 (P1); (**b**) Cumulative index (black line) and detrended cumulative sum (red line) of D PRIFT index of sequence 1. The blue line corresponds to the linear trend or drift of the cumulative sum of D PRIFT index.

**Figure 3 entropy-21-00852-f003:**
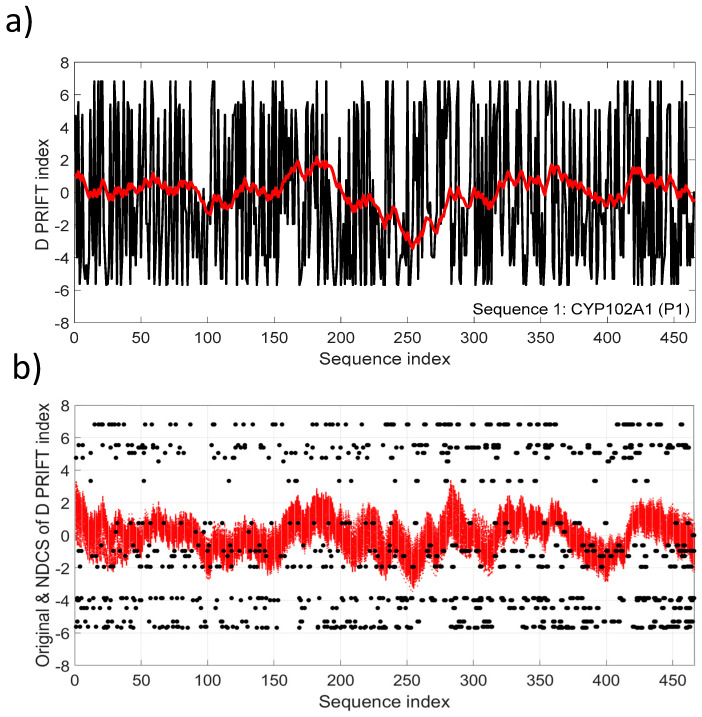
(**a**) D PRIFT index of sequence 1, which is parent CYP102A1 (P1). A superimposed red line corresponds to the normalized detrended cumulative sum (NDCS) of D PRIFT index; (**b**) Original (black dot) and normalized detrended cumulative sum (small red cross) of D PRIFT index for 242 protein sequences.

**Figure 4 entropy-21-00852-f004:**
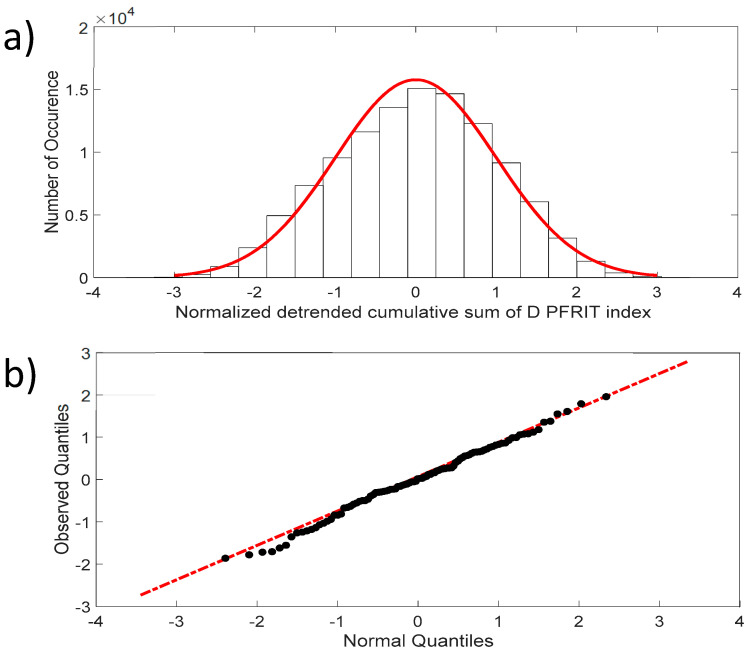
(**a**) Distribution of the NDCS of D PRIFT index changes for all sequences (black dots). Red line corresponds to Gaussian distribution; (**b**) QQ-plot of the NDCS of D PRIFT index changes quantiles and Gaussian quantiles. Red dotted line is a linear fitting of observed quantile distribution versus normal quantile distribution.

**Figure 5 entropy-21-00852-f005:**
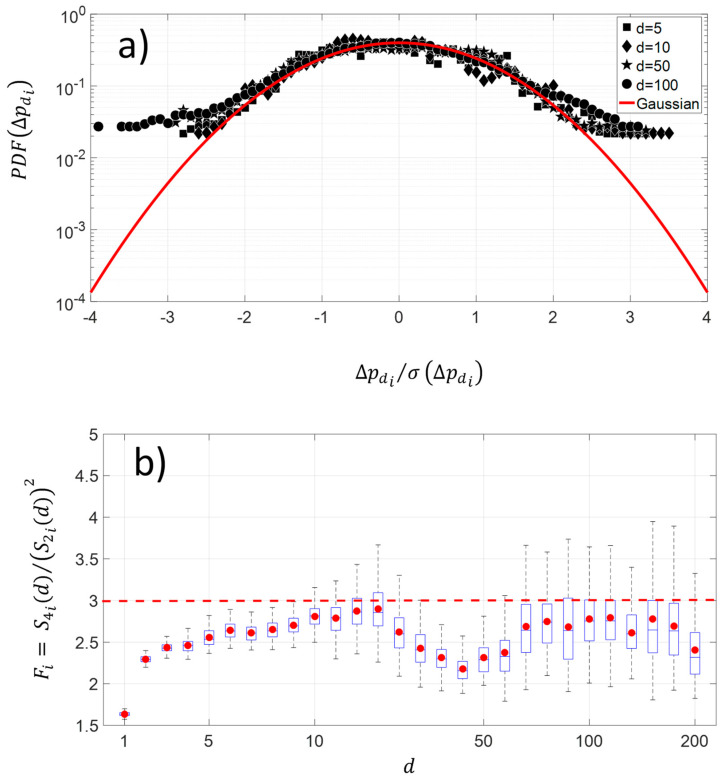
(**a**) Shape of average and normalized experimental probability functions (PDFs) of the increment of NDCS of PRIFT index changes at different distances in pairwise sequence d=5, d=10, d=50, and d=100 of 242 protein sequences. (**b**) Deviation of NDCS of PRIFT index changes distribution with respect to the Gaussian distribution at different pairwise sequence d.

**Figure 6 entropy-21-00852-f006:**
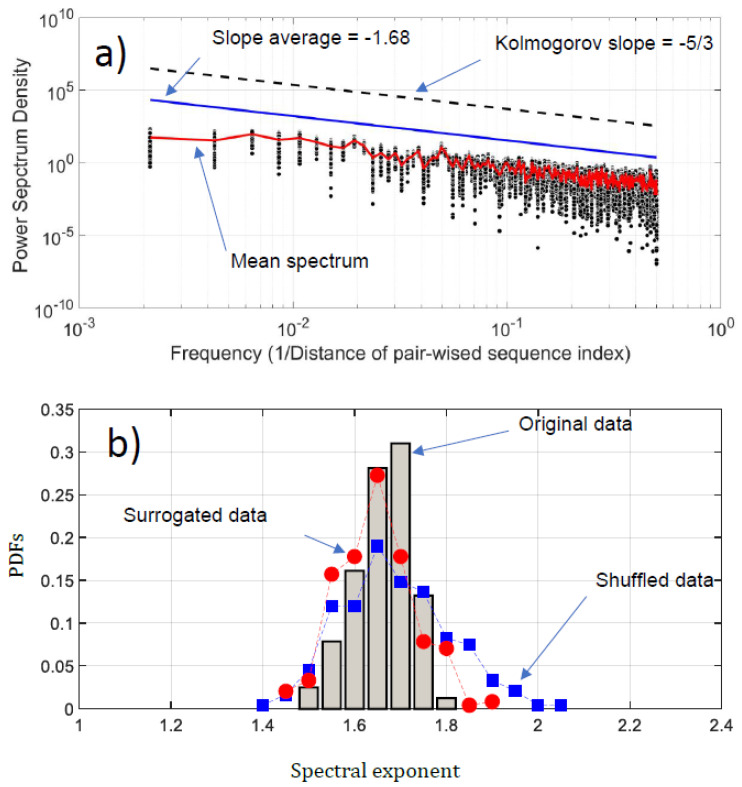
(**a**) Power spectrum density (PSD) of the NDCS of D PRIFT index changes of all 242 protein sequences $S_{i}$ (black dots); (**b**) PDFs of the spectral exponent estimated from Fourier analysis. We have superimposed the PDFs obtained with surrogated (red spots) and shuffled data (blue squares).

**Figure 7 entropy-21-00852-f007:**
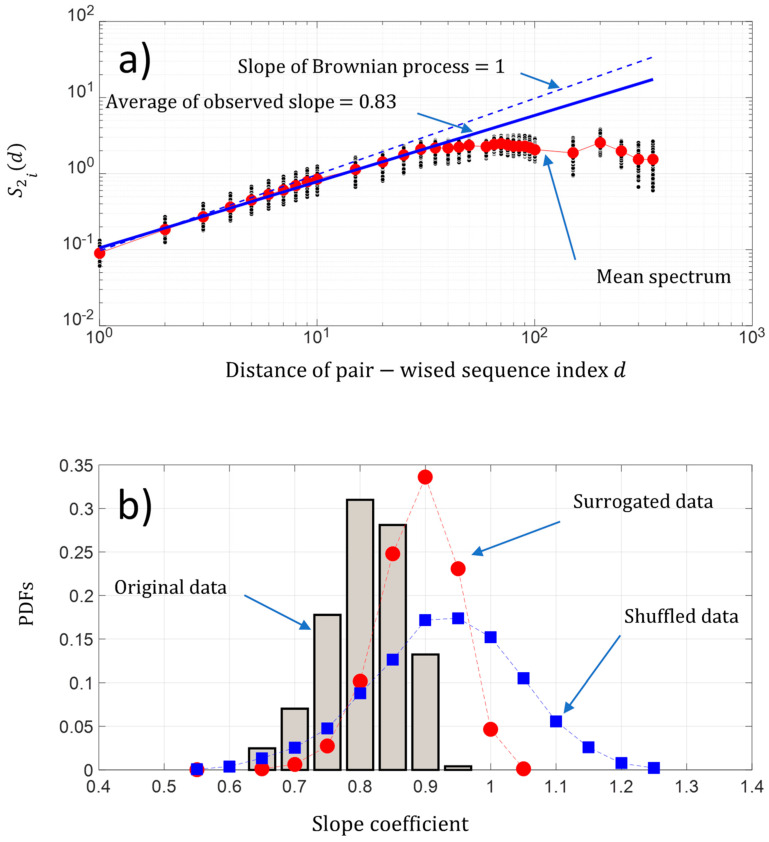
(**a**) Log-log presentation of the second-order moment ${S_{2}}_{i}\left(d\right)$ of the NDCS of D PRIFT index changes of all 242 protein sequences ${S_{2}}_{i}$ versus the distance d of the pairwise protein sequence index (black dots); (**b**) PDFs of the slope of the scaling law distribution of the second-order moment ${S_{2}}_{i}\left(d\right)$ of the NDCS of D PRIFT index changes estimated for each protein sequence $S_{i}$. We have superimposed the PDFs obtained with surrogated (red spots) and shuffled data (blue squares).

**Figure 8 entropy-21-00852-f008:**
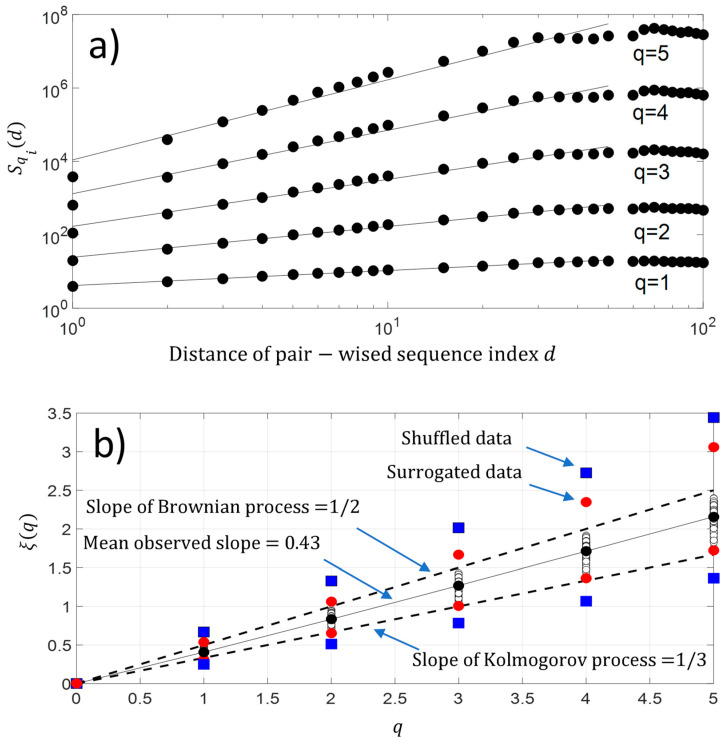
(**a**) Experimental high-order structure functions ${S_{q}}_{i}\left(d\right)$ with varying moments for q=1, 2, 3, 4, and 5; (**b**) Generalized Hurst exponent ξ(q). We have added the maximum and minimum value of ξ(q) obtained with surrogated and shuffled data.

**Figure 9 entropy-21-00852-f009:**
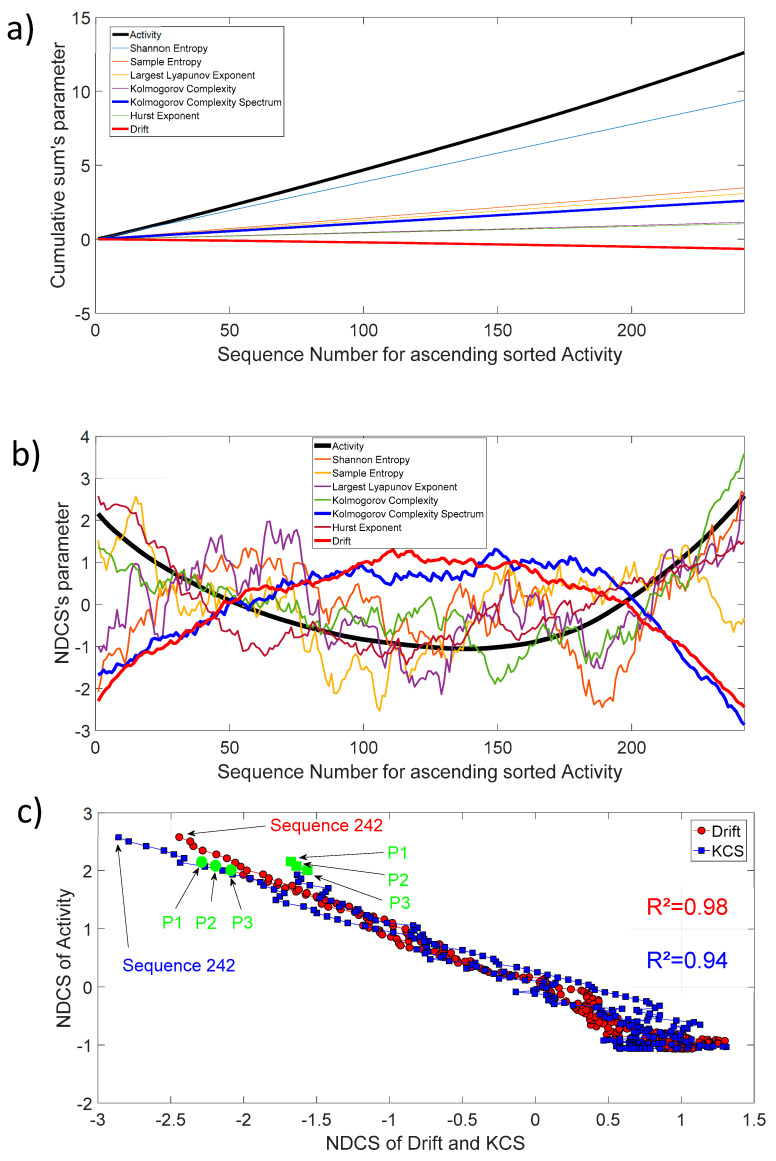
(**a**) Cumulative sum of activity, entropy, chaos, complexity, fractal, and drift parameters for ascending sorted activity; (**b**) Normalized detrended cumulative sum of activity, entropy, chaos, complexity, fractal, and drift parameters for ascending sorted activity; (**c**) NDCS of activity (ACT) versus NDCS of drift (DRF) and Kolmogorov complexity spectrum (KCS). The square of the correlation coefficient $R^{2}$ for both curves is added on the figure. The first and last sequence positions of the 242 ordered sequences are also shown. The green circle and square symbol indicate the position of the parents CYP102A1 (P1), CYP102A2 (P2), and CYP102A3 (P3) in this diagram.

**Figure 10 entropy-21-00852-f010:**
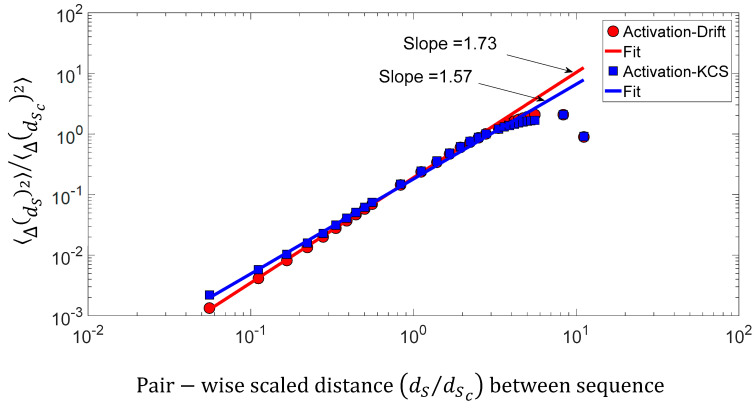
Log–log presentation of the mean square displacement $\langle \Delta {\left(d_{S}\right)}^{2}\rangle /\langle \Delta {\left({d_{S}}_{c}\right)}^{2}\rangle$ of the bivariate (KCS, ACT) parameter versus the pairwise scaled d distance $\left(d_{S}/{d_{S}}_{c}\right)$ between sequences.

**Table 1 entropy-21-00852-t001:** Conversion rule of protein sequence of AA index 532—D PRIFT index [30].

AA Index 532 D PRIFT Index (Cornette et al. 1987)																				
Letter	A	C	D	E	F	G	H	I	K	L	M	N	P	Q	R	S	T	V	W	Y
Value Index	−5.68	−5.62	−5.30	−4.47	−3.99	−3.86	−1.94	−1.92	−1.28	0.96	0.62	0.21	0.75	3.34	4.54	4.76	5.06	5.39	5.54	6.81

**Table 2 entropy-21-00852-t002:** Descriptive statistics of entropy, chaos, and complexity of the NDCS of D PRIFT index changes for 242 protein sequences.

D PRIFT Index		Entropy		Chaos	Complexity		Fractal
		Information	Regularity				
NDCS Data		Shannon Entropy	Sample Entropy	Largest Lyapunov Exponent	Kolmogorov Complexity	Kolmogorov Complexity Spectrum	Hurst Exponent
Minimum	Original	3.671	1.251	0.930	0.247	1.008	0.347
	Surrogate	3.514	1.051	0.730	0.152	1.046	0.332
	Shuffled	3.498	0.600	0.332	0.095	1.046	0.273
Mean	Original	3.880	1.433	1.277	0.475	1.071	0.432
	Surrogate	3.875	1.289	1.070	0.399	1.105	0.481
	Shuffled	3.911	1.147	0.911	0.328	1.103	0.498
Median	Original	3.888	1.436	1.286	0.475	1.065	0.436
	Surrogate	3.895	1.296	1.072	0.399	1.103	0.482
	Shuffled	3.933	1.154	0.906	0.323	1.103	0.498
Maximum	Original	4.066	1.618	1.601	0.647	1.141	0.481
	Surrogate	4.131	1.547	1.501	0.646	1.179	0.615
	Shuffled	4.188	1.604	1.469	0.627	1.160	0.690
Standard deviation	Original	0.084	0.063	0.117	0.084	0.031	0.027
	Surrogate	0.117	0.094	0.143	0.081	0.023	0.033
	Shuffled	0.130	0.188	0.220	0.109	0.022	0.058
1st quartile	Original	3.833	1.389	1.207	0.418	1.046	0.420
	Surrogate	3.805	1.226	0.969	0.342	1.084	0.459
	Shuffled	3.842	1.017	0.750	0.228	1.084	0.459
3rd quartile	Original	3.940	1.470	1.351	0.533	1.103	0.450
	Surrogate	3.963	1.355	1.160	0.456	1.122	0.503
	Shuffled	4.005	1.297	1.045	0.399	1.122	0.538

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
