# Peer review of "Non-Linear Dynamics Analysis of Protein Sequences. Application to CYP450"

_entropy, 2019, doi:10.3390/e21090852_

Round 1

Reviewer 1 Report

The authors present a new approach to address the non-stationary and nonlinear fluctuation of changes encountered in protein sequence. In a more detailed way, it searches to answer the following research guide questions:

“Can statistical, nonlinear and complexity analysis give us some information about the pattern in a protein sequence and its changes along the sequence and also the next, or other sequences? Can we group sequences according to their activity but also to their morphological pattern?”

The paper is really very well organized and written: the problem is scientifically relevant and interesting. The methodology is well designed and the definition of the validation approach is adequate: the results seem coherente with the proposed methodology objectives.

The only comment that I wish to make about what could be improved is referred to the introduction. Within this section the analysis of state of the art and background is included, but i find that this part suffers from some weakness: we dont know other approaches to the problem, or any related work. I find that the references that are cited are more related to the relevance of the problem but they dont inform in a detailed way about what a state of the art section must report about. I think that more background and context information must be provided

Author Response

Thanks for the comments of reviewer 1. Please find our reply as attached file.

Best regards

Pr. M. BESSAFI  

Reviewer 2 Report

The authors of the article entitled "Non-linear dynamics analysis of protein sequences" describe a formal approach to evaluate the non-stationary and non-linear fluctuation of changes in protein sequences.

In order to perform this study, the authors employ numerical values obtained from AAindex database to transcribe amino acid sequences to numeric values allowing the application of distinct methods such as algorithmic information and statistical analysis to estimate change fluctuations and complexity in proteins sequences.

Notwithstanding this study describes a detailed and comprehensive application of well-established formal concepts to estimate non-linearity, it lacks well grounded premises that are fundamental to justify the development of the entire study. Some of these points are described below:

How representative is the protein cytochrome P450 (CYP450) and its 242 recombinant derivatives used in this study to depict the encoding pattern of protein sequences in general without a severe bias? For comparison, as on September 2018, there was ~18k proteins of known distinct families only in Pfam database. When restricting the study of protein sequence non-linearity only for CYP450 derivatives, we still have a point to consider for the selected ensemble. The chimeric constructions of CYP450 are generated by combining fixed fragments as building blocks from parent proteins CYP450 A1, A2 and A3. How this discretization affects the complexity and non-linearity dynamics obtained by statistical and information theory analysis? An approximation to a random generator of variants for stability selection seems to be flimsy.  The authors use DPRIFT index from AAindex database to distinguish 20 standard amino acids by their hydrophobicity values. This description is oversimplified and does not account for many other properties of amino acids that are well known to strongly affect pattern changes in protein sequences along families, such as volume, aromaticity and different charge states for the same amino acid in distinct positions (which could be estimated from native CYP450 fold). Maybe these variables could be considered with an extended AAindex catalog. All analysis developed in the article do not differentiate amino acids that are on the central region of the sequence and on its extremes (close to N-terminal or C-terminal). It is expected a key influence of aminoacid position in primary structure for thermal stability, which was a criteria to select CYP450 constructions in the study.

Finally, the authors present no discussion or interpretation about the meanings of the results found and their implications to the dynamics of changes in protein sequences. Rather, they only provide a description of the values found. The article would be much more enlightening with some interpretations of the results found, e.g. what are the biological effects expected due to the agreement of sequence fluctuations with Kolmogorov power-law, the monofractal behaviour and detrended curves. The only statement in conclusion: "there is some predictability of protein changes" is very vague and does not really contribute to the field. 

Moreover a few points are worthy to be mentioned:

 - Since the exposure of continuous amino acids sequences to solvent or their occlusion in protein cores is a fundamental requirement for proteins to fold in functional arrangements, distribution of hidrophobic and polar amino acids should be inhomogeneous along the entire system and rather concentrated in narrow ranges. The influence of this effect to the derivated sequences should be discussed in the paper when studying non-linearity. Also, these distributions could be also included in Figure 1 when showing the distribution of D PRIFT index.

 - Moreover, a good track for comparison of derivative sequences would be to include labels to identify the positions of parent CYPs in all analysis (in special Figure 9).

Some few typos to correct:

line 149: asSi -> as Si

line150: reference 22 follows a distinct pattern.

line 228: [5.68 6.81] -> [5.68-6.81]

Figure 1 y-axis label: D RPIFT -> D PRIFT 

378: Thermostablity -> Thermostability

393: compelxity -> complexity

394: asascending -> as ascending

402: bewteen -> between

Author Response

(The authors gave the same response as above.)

Round 2

Reviewer 1 Report

I have read the comments and the explanations and changes are correct

Reviewer 2 Report

The authors provided detailed and adequate answers to all questions raised during revision. I believe the results presented in this study add a valuable contribution to the field and promote new studies towards a more deterministic understanding of structural biology through the point of view of information theory.

Nevertheless, I believe the title and abstract chosen for this paper are too bold and could give the idea that the results depict a general behaviour of protein sequences. Therefore I suggest the authors to clearly reference protein cytochrome P450 system as the scope of these studies (e.g. a title "Non-linear dynamics analysis of CYP450 protein sequences").

Setting aside these minor changes, I gladly commend the paper for publication in Entropy.

Author Response

Thanks for the valuable comments of the reviewer. 

We have changed the title of the manuscript and suggest “Non-linear dynamics analysis of protein sequences. Application to CYP450.”

Round 3

Reviewer 2 Report

The authors provided detailed and adequate answers to all questions raised during the entire revision process. I believe the results presented in this study add a valuable contribution to the field and promote new studies towards a more deterministic understanding of structural biology through the point of view of information theory. Therefore, I gladly commend this paper for publication in Entropy.